# Assessment of Glycemic Response to Model Breakfasts Varying in Glycemic Index (GI) in 5–7-Year-Old School Children

**DOI:** 10.3390/nu13124246

**Published:** 2021-11-26

**Authors:** Sandra I. Sünram-Lea, Gertrude Gentile-Rapinett, Katherine Macé, Andreas Rytz

**Affiliations:** 1Department of Psychology, Lancaster University, Lancaster LA1 4YF, UK; 2Nestlé Research Center, 1000 Lausanne, Switzerland; gertrude.gentile-rapinett@hh.global (G.G.-R.); catherine.mace@rdls.nestle.com (K.M.); andreas.rytz@rdls.nestle.com (A.R.)

**Keywords:** post-prandial glycemic response (PPGR), cognition, children, Glycemic Index (GI)

## Abstract

Reduced Glycemic Index (GI) of breakfast has been linked to improved cognitive performance in both children and adult populations across the morning. However, few studies have profiled the post-prandial glycemic response (PPGR) in younger children. The aim of this study was to assess PPGR to breakfast interventions differing in GI in healthy children aged 5–7 years. Eleven subjects completed an open-label, randomized, cross-over trial, receiving three equicaloric test beverages (260 kcal) consisting of 125 mL semi-skimmed milk and 50 g sugar (either glucose, sucrose, or isomaltulose). On a fourth occasion, the sucrose beverage was delivered as intermittent supply. PPGR was measured over 180 min using Continuous Glucose Monitoring (CGM). The incremental area under the curve (3h-iAUC) was highest for the glucose beverage, followed by intermittent sucrose (−21%, *p* = 0.288), sucrose (−27%, *p* = 0.139), and isomaltulose (−48%, *p* = 0.018). The isomaltulose beverage induced the smallest Cmax (7.8 mmol/L vs. >9.2 mmol/L for others) and the longest duration with moderate glucose level, between baseline value and 7.8 mmol/L (150 vs. <115 min for others). These results confirm that substituting mid-high GI sugars (e.g., sucrose and glucose) with low GI sugars (e.g., isomaltulose) during breakfast are a viable strategy for sustained energy release and glycemic response during the morning even in younger children.

## 1. Introduction

Due to its association with healthier macro- and micronutrient intake, body mass index, and lifestyle, habitual breakfast consumption has been recommended as part of a healthy diet [1]. Breakfast has also been promoted to improve cognitive function and academic performance, leading to the provision of breakfast initiatives in schools. Although there is debate about the degree to which breakfast consumption is key to optimizing various health outcomes across different populations, children may be particularly sensitive to the nutritional effects of breakfast on brain activity and associated cognitive outcomes. The reason for their greater susceptibility is likely to be due to greater energetic needs during this period compared to adults [2]. Childhood is a time of intense learning and children learn many basic concepts in reading, writing, and arithmetic during these years. Consequently, the high rate of glucose utilization, which is maintained from age 4 to 10 years [3] coincides with a period of one of the most metabolically expensive cognitive processes. In order to maintain this higher metabolic rate, a continuous supply of energy derived from glucose is needed, hence breakfast consumption may be vital to providing adequate energy supply for children, especially as food intake ad libitum is not possible or is at least limited during school mornings. 

There is mechanistic evidence linking the post-prandial glycemic response (PPGR) to cognitive performance in both children and adult populations [4]. Therefore, when considering what type of breakfast may be most beneficial, the Glycemic Index (GI), which summarizes the rate at which food increases and maintains blood glucose, appears to be an important modulating factor. After a high GI meal, plasma glucose concentrations rise rapidly, causing a high peak glucose level and a concomitant high insulin response, resulting in a rapid blood glucose disposal, which in turn may cause blood glucose levels to decrease to below the fasting concentration in the later post-prandial period [5]. Low-GI foods result in more moderate peak blood glucose increments and may also maintain a prolonged net increment in blood glucose above the fasting concentration. A recent meta-analysis confirmed that in adults, low GI breakfast interventions significantly lowered glycemic response 60, 90, and 120 min post-consumption, with a similar trend observed for low glycemic load [6]. In addition to the importance of an adequate glucose supply to the brain, a PPGR characterized by a low peak but a sustained net increment of glucose above fasting concentrations may provide benefits in cognitive functioning during the post-prandial phase. Implicit in the recommendation to formalize GI as a dietary guidance tool is the assumption that the PPGR an individual has to a given food is solely due to intrinsic property of the food consumed and similar among individuals regardless of metabolic and physiological factors. However, there is increasing evidence of high inter-individual variability in PPGR to the same foods [7]. Few studies have profiled PPGR to foods in children [8]. However, based on the interpersonal variability, the general assumption that responses in healthy pediatric populations should be roughly the same as in adults needs to be tested. Moreover, although it is assumed that low GI interventions provide a stable glucose supply, the evidence is still equivocal, and the concept remains to be clearly substantiated. 

Over the last four decades, technical advances have led to the development of minimally invasive continuous glucose monitoring (CGM) devices. Such devices rely on the measurement of glucose concentrations in the interstitial fluids, which offers numerous advantages considering that subcutaneous tissue is easily accessible for sensor implantation. The minimal invasiveness of CGM devices compared to traditional intravenous or capillary glucose monitoring means that glucose can be monitored in real-time for longer periods and allows for a more comprehensive glycemic assessment. One such device, the FreeStyle Libre is recommended and approved for use in adults and children older than 4 years-old with type 1 or type 2 diabetes. Previous research has shown that interstitial glucose measured with the device is well correlated with capillary blood glucose in Type I and Type II adult diabetic patients [9] and Type I diabetic children [10]. Moreover, the advantage of the FreeStyle Libre is that this time-lag is very short (around 4.5 min) compared to other devices, for which time-lags have been reported ranging from 5 to 15 min, extending up to 45 min [11]. 

The clinical safety and accuracy of the device within a diabetic pediatric population aged 4–17 years has been demonstrated [12]. The FreeStyle device has not been tested in healthy children to measure glycemic excursion in response to a meal or food products. However, other CGM devices have been shown to detect differences in PPGR after consumption of meals/products with differing carbohydrate quantity or quality in normoglycemic populations. A continuous glucose monitoring system (Medtronic MiniMed CGMS; Northridge, CA, USA) was used to assess PPGR to three different GI interventions (glucose drink: GI 100, GL 65; a full milk beverage: GI 27, GL 5; and a half milk/glucose beverage: GI 84, GL 35) in children aged 10–12 years old by Brindal et al. [8]. They found that glucose measures obtained through CGM demonstrated significant differences for each meal condition.

Understanding the potential influence of nutrition and, more specifically, breakfast interventions on children’s cognitive function remains a high priority, given its application to learning and achievement at school. However, the differences in PPGR after consumption of products with differing carbohydrate quantity or quality need to be established in that age group. Consequently, the purpose of this study was the assessment of PPGR to breakfast interventions with low, moderate and high-GI, including the later phase (>120 min), which is beyond that traditionally used for GI calculation.

## 2. Materials and Methods

### 2.1. Samples

The study compared 3 model breakfasts that were 3 beverages consisting of 50 g sugar (either glucose, isomaltulose, or sucrose) dissolved in 125 mL commercial semi-skimmed milk. The 3 beverages were equicaloric (260 kcal), and their simple composition allowed estimating their Glycemic Index (eGI) and Glucose Load (eGL) [13]. The eGI was classified as high (>70) for the glucose beverage, mid (between 55 and 70) for the sucrose beverage, and low (<55) for the isomaltulose beverage (Table 1).

### 2.2. In-Vivo Study

A total of 11 healthy children (8 females, 3 males; aged 5−7 years with mean = 6.5 year and SD = 0.7 year; BMI mean = 16.2 and SD = 1.0 kg/m^2;^ BMI percentile range observed in sample 39th−88th percentile) were studied on 4 occasions, after a 12 h overnight fasting that followed a standardized evening meal consisting of pizza, mozzarella salad, and fruit sticks (460 kcal, providing 25–30% of daily energy intake, and low amounts of slowly digestible carbohydrates). Fasting interstitial glucose flash readings were obtained 5 min before (T−5) and at the time of beverage intake (T0). These 2 values were averaged to serve as baseline glucose value. After baseline value was obtained, subjects consumed, within 12 min, one of the 3 test beverages containing 50 g of either glucose, isomaltulose, or sucrose. On a fourth occasion, the sucrose beverage was delivered as intermittent supply with one third being delivered at T0, one third at T30, and the final third at T60. From T0, interstitial glucose was measured continuously every 15 min until T180. During this period, subjects could watch a movie and/or play computer or board games, draw or read. After the final glucose reading had been obtained at T180, children were offered a snack providing approximately 100 kcal; they could choose either spreadable cheese with breadsticks or apple slices and semi−skimmed milk.

The study followed a single-center, randomized, open-label, cross-over design, with subjects being randomly allocated to a sequence of the 4 tested conditions using a Williams Latin square design that counterbalanced position and first-order carry-over effects [14]. To comply with the 12 h overnight fasting, children who were due to present at the testing unit at 08:00 were asked to complete their dinner by 20:00 the previous evening. In line with previous research, the time window between 2 consecutive test visits was 24 to 48 h [8].

Before the 4 test visits, children agreed to participate in the research project through written assent, and their parents/caregivers signed an informed consent declaration after getting all information about the study. Two days before the first testing visit, a nurse attached the CGM sensor (FreeStyle Libre supplied by Abbott Laboratories Ltd., Abbott House, Vanwall Business Park, Vanwall Road, Maidenhead, Berkshire, SL6 4XE, UK) on the back of the arm of the children to allow for a 24 h measurement stabilization [9]. Tegaderm (transparent medical dressing) was used to ensure that the sensor stayed in place. Parents/caregivers were instructed on the use of the device and were given contact details in case of concerns or an adverse event. At the end of the final test session, the nurse removed the sensor.

The study served as a pilot for a potential bigger study combining continuous glucose monitoring with cognitive outcomes. The sample size was, therefore, kept small on purpose while complying with standard protocols [15].

The study protocol was reviewed and approved by the Ethics Committee of the NHS Health Research Authority. The information sessions were held at the Infancy and Early Development Research Unit at Lancaster University, and the study was conducted at the Clinical Research Facility of the Royal Preston Hospital.

### 2.3. Data Analysis

The primary endpoint of this study was the 3-h incremental area under the curve (3h-iAUC) of post-prandial glucose response. This 3h-iAUC was estimated using the trapezoid method on each individual curve. Secondary endpoints derived from the post-prandial glucose response were 2h-iAUC, maximal glucose value (Cmax), maximal incremental glucose value (iCmax), the time to reach this value (Tmax), the duration with glucose > 7.8 mmol/L (D-high), the duration with glucose < baseline (D-low), and the duration with glucose in-between (D-moderate) and all cross-sectional timepoints, every 15 min between T0 and T180. Mean glucose response curves are shown in a graph using mean and standard error (SE) at each cross-sectional time-point (Figure 1). 

Inference was based on the per-protocol set. Data were excluded for one visit of one subject (illness) and for one visit of another subject (administration of Calpol prior to session). Endpoints derived from glucose curves were tabulated using Mean ± SE and *p*-values associated with the paired *t*-test vs. the glucose beverage, with 2-sided 5% significance level. A sensitivity analysis was performed using a mixed model to impute missing data and to consider potential systematic position or carry-over effects [16]. Since none of these effects were close to reaching statistical significance, these analyses were not further presented.

## 3. Results

### 3.1. 3h-PPGR and Incremental Area under the Curve 

Average 3h-PPGR curves show that the fasting baseline value was 5.4 ± 0.17 mmol/L (mean ± SE, *N* = 11) and that all three test beverages peaked between 30 and 45 min before decreasing rapidly until 60 min. The decrease was then attenuated between 60 and 180 min, reaching an average baseline between 150 and 180 min (Figure 1). The mean 3h-PPGR curve appears to be highest for glucose during the whole duration, followed by sucrose (first 105 min) and isomaltulose, which is higher than sucrose for the last 60 min. 

These curves translate into the highest 3h-iAUC for the glucose beverage (284 ± 58 mmol/L*min), followed by sucrose (27% decrease, 206 ± 58, *p* = 0.139) and isomaltulose (48% decrease, 147 ± 21, *p* = 0.018). These relative differences are already present after 2 h (Table 2). Glucose shows highest 2h-iAUC (232 ± 45 mmol/L*min), followed by sucrose (21% decrease, 184 ± 29, *p* = 0.218) and isomaltulose (50% decrease, 116 ± 18, *p* = 0.012). With *N* = 11 subjects, only the difference between glucose and isomaltulose appears, therefore, to be statistically significant.

The intermittent supply of the sucrose beverage leads to slightly higher iAUC values than the single supply of the same beverage, both after 3 h (224 ± 40 mmol/L*min) and after 2h (200 ± 35 mmol/L*min). It is not significantly different from sucrose or glucose.

### 3.2. Secondary Endpoints 

The post-prandial glycemic peak was highest for glucose (9.7 ± 0.61 mmol/L), followed by sucrose (9.3 ± 0.49, *p* = 0.501), intermittent sucrose (9.2 ± 0.45, *p* = 0.497), and isomaltulose (7.8 ± 0.27, *p* = 0.005). In terms of incremental Cmax, this translates into highest iCmax for glucose (4.3 ± 0.62 mmol/L), followed by sucrose (10% decrease, 3.9 ± 0.45, *p* = 0.488), intermittent sucrose (12% decrease, 3.8 ± 0.43, *p* = 0.503), and isomaltulose (45% decrease, 2.4 ± 0.31, *p* = 0.006). As for the iAUC, with *N* = 11 subjects, only the difference between glucose and isomaltulose appears to be statistically significant.

The glucose peak appears in average after, respectively, 37 ± 5 min for glucose, 40 ± 7 for sucrose (37 ± 3 for intermittant sucrose), and 48 ± 11 for isomaltulose. This small delay in time observed in isomlatulose (+10 min) was not statistically significant with *N* = 11 subjects.

The average duration with moderate glucose value (between baseline value and 7.8 mmol/L) was longest after ingestion of isomaltulose (150 ± 10 min) followed by glucose (115 ± 13), intermittent sucrose (110 ± 10), and sucrose (97 ± 10). Only the almost 1-h difference between isomlatulose and sucrose was statistically significant (*p* = 0.008). Iomaltulose is the beverage that induces shortest duration with glucose below baseline (23 ± 9 min) and shortest duration with glucose higher than 7.8 mmol/L (7 ± 3 min, significantly lower than glucose, 41 ± 14, *p* = 0.039).

## 4. Discussion

The three test beverages consisting of 50 g sugar (either glucose, sucrose, or isomaltulose) dissolved in 125 mL semi-skimmed milk lead to different PPGR in 5–7-year-old children. The glucose beverage is most glycemic, followed by sucrose (27% reduction in 3 h-iAUC and 0.4 mmol/L reduction in Cmax), and isomaltulose (48% reduction in 3h-iAUC and 1.9 mmol/L reduction in Cmax). The order of magnitude of these reductions vs. glucose observed in 5–7-years-old children are in-line with those observed in adults; they are only slightly lower than the predicted eGL reductions that were, respectively, 36% for sucrose and 64% for isomaltulose (Table 1). In addition, the glucose beverage, that has a predicted eGL of 49.6 g, induces an average 2h-iAUC of 232 ± 45 mmol/L*min in 5–7-years-old children. This is very close to the iAUC induced in adults by 50 g glucose diluted in water, as shown by diverse studies. For the 140 adults with mean age of 29.6 years underlying the eGL predictions [13], this iAUC was 237 mmol/L*min on average. It was also reported to be 216 mmol/L*min in a study with 20 adults of mean age 21.9 years [17].

The results showed that using a CGM system in children aged 5–7 years resulted in significant differences between three test beverages classified as low, mid, or high GI. The observed differences were comparable to those observed in adults. Beverages featuring 50 g sucrose or glucose were associated with a greater post-apex fall across the morning and 50 g isomaltulose with a lower PPGR. Evidence for an association between a lower PPGR and better cognitive performance across the morning in children is inconclusive. However, there is some evidence to suggest a positive effect on delayed memory performance following low-GI breakfasts [18].

The isomaltulose beverage induced the smallest Cmax (7.8 mmol/L vs. >9.2 mmol/L for others) and the longest duration with moderate glucose level, between baseline value and 7.8 mmol/L (150 vs. <115 min for others). These results suggest that substituting mid-high GI sugars (e.g., sucrose and glucose) with low GI sugars (e.g., isomaltulose) during breakfast could be a viable strategy for sustained cognitive performance during the morning. 

The intermittent supply of the sucrose beverage, which was designed to increase the duration of continuous glucose supply, did not minimize oscillations in glucose levels and did, therefore, not mimic the low GI intervention. This result suggests that the intrinsic quality of the breakfast was more important than the feeding pattern and that a unique breakfast supply before school was appropriate.

This pilot study used a cross-over design that helped keep the sample size as small as possible to compare PPGR induced by different test breakfasts. Considering the primary outcome (3h-iAUC), the sample size of *N* = 11 allowed to significantly discriminate glucose from isomaltulose (48% reduction) but not from sucrose (27% reduction). To detect this latter effect with significance level of 5% and power of 80%, the sample size should be *N* = 30 in a cross-over setup and *N* = 50 in a parallel setup. These results help to properly power future studies aiming at understanding the potential influence of PPGR induced by breakfast on children’s cognitive function. This remains a high priority, given its application to learning and achievement at school.

## Figures and Tables

**Figure 1 nutrients-13-04246-f001:**
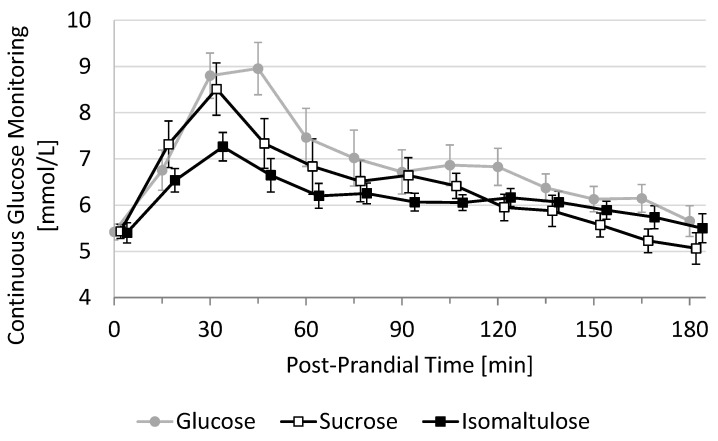
Average 3h-PPGR of the three test beverages featuring 50 g sugar dissolved in 125 mL semi-skimmed milk. Data are shown as Mean ± SE at cross-sectional time-points every 15 min (with *N* = 11 subjects).

**Table 1 nutrients-13-04246-t001:** Three test beverages featuring 50 g sugar dissolved in 125 mL semi-skimmed milk: glycemic Index (GI) of the sugars estimated GI of the beverage (eGI) and estimated Glycemic Load of a serving (eGL).

Beverage	GI Sugar	eGI Beverage	eGL Serving
Glucose	100	89	49.6 g
Sucrose	62	57	31.8 g (64%) ^1^
Isomaltulose	32	32	17.7 g (36%)

^1^ expressed relative to eGL of the Glucose beverage.

**Table 2 nutrients-13-04246-t002:** Descriptive statistics (mean ± SE, *N* = 11) for the three beverages featuring 50 g sugar dissolved in 125 mL semi-skimmed milk, and the intermittent sucrose beverage supply. For pairwise comparisons vs. glucose (paired *t*-test, two-sided), *p*-values are given in brackets.

Endpoint	Glucose	Sucrose	Isomaltulose	Sucrose 3′
3h-iAUC (mmol/L*m)	284 ± 58	206 ± 35 (*p* = 0.139)	147 ± 21 (*p* = 0.018)	224 ± 40 (*p* = 0.288)
2h-iAUC (mmol/L*m)	232 ± 45	184 ± 29 (*p* = 0.218)	116 ± 18 (*p* = 0.012)	200 ± 35 (*p* = 0.488)
Cmax (mmol/L)	9.7 ± 0.61	9.3 ± 0.49 (*p* = 0.501)	7.8 ± 0.27 (*p* = 0.005)	9.2 ± 0.45 (*p* = 0.497)
iCmax (mmol/L)	4.3 ± 0.62	3.9 ± 0.45 (*p* = 0.488)	2.4 ± 0.31 (*p* = 0.006)	3.8 ± 0.43 (*p* = 0.503)
Tmax (min)	37 ± 5	40 ± 7 (*p* = 0.756)	48 ± 11 (*p* = 0.377)	37 ± 3 (*p* = 1.000)
D-high (min)	41 ± 14	33 ± 8 (*p* = 0.449)	7 ± 3 (*p* = 0.039)	31 ± 11 (*p* = 0.387)
D-low (min)	25 ± 8	50 ± 12 (*p* = 0.173)	23 ± 9 (*p* = 0.887)	38 ± 10 (*p* = 0.341)
D-moderate (min)	115 ± 13	97 ± 10 (*p* = 0.350)	150 ± 10 (*p* = 0.066)	110 ± 10 (*p* = 0.720)

## Data Availability

The data presented in this study are available on request from the corresponding author. The data are not publicly available due to intellectual property rights.

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
