# Peer review of "Assessment of Glycemic Response to Model Breakfasts Varying in Glycemic Index (GI) in 5–7-Year-Old School Children"

_nutrients, 2021, doi:10.3390/nu13124246_

Round 1

Reviewer 1 Report

This is a paper related to GI and breakfast.

This is a well known theme, with large body of evidence demonstrating that regular breakfast consumption is associated with a variety of nutritional and lifestyle-related health outcomes in large diverse samples of young people. There is emerging evidence in young people that suggests certain breakfasts are particularly beneficial for health, with much of this evidence focusing on ready-to-eat cereals and breakfast glycaemic index (GI). Substituting a high GI (HGI) breakfast for a low GI (LGI) breakfast has already been described as particularly beneficial for overweight young people through increased glycaemic control and satiety. Furthermore, some meta-analytic results revealed that Low GI breakfasts significantly reduced postprandial blood glucose concentrations at time points 60 and 120 min. Similar trends were also indicated following the stratification of studies according to the glycemic load, showing a more pronounced decline in glycemic response among individuals with metabolic impairments. These results highlight the benefits of lowering breakfast meal GI to provide clinically relevant reductions in acute glucose response.

As the glycemic index is a value assigned to foods based on how slowly or how quickly those foods cause increases in blood glucose levels, results shown in this paper are well expected, thus do not add (to my humble opinion) any very strong insights about proposed topic of the paper.

Author Response

We would like to thank the reviewer for taking the time to comment on our manuscript and for making valuable suggestions. We fully agree with the reviewer’s assertion that a lower glycaemic profile following low GI and also low GL interventions has been repeatedly demonstrated in adult populations and confirmed by a recent meta-analysis of randomized controlled trials (Toh, Koh, & Kim, 2020). We have added a reference for the recent meta-analysis (Page 2 Line 56-59). However, the evidence base for healthy paediatric populations is limited and combined with reports of high inter-individual variability in glycaemic responses (Venn & Green, 2007, Vega-López et al., 2007; Vrolix, Mensink, 2010, Zeevi, Korem, Zmora et al., 2015) we feel that confirmation that responses in healthy paediatric populations follow the same pattern adds to our knowledge base.

Reviewer 2 Report

The authors investigated the post-prandial glycemic response to varying glycemic index foods that were isocaloric in 11 healthy children after a standardized evening meal and fast using an open label randomized crossover design.

Methods: The authors utilized three different types of carbohydrate beverages and a FreeSTyle Libre CGM system to measure the 3hr response in the participants. Although BMI is described, it would be helpful to know the BMI percentile of the participants since that is recommended to be used in children. The window between two consecutive tests was 24 to 48 hours. (Line 132) It would be helpful to cite any relevant literature that supports this interval is adequate for washout of previous treatment.

Results: The authors discuss results for the 3h-iAUC differences for the three beverages: " glucose beverage (284±58 182 mmol/L*min), followed by sucrose (27% decrease, 206±58, p=0.139) and isomaltulose (48% 183 decrease, 147±21, p-value=0.018)" and similarly for 2h-iAUC, as well as other important secondary end points. However, there are no cognitive outcomes reported although authors acknowledge that those might be reported in a later bigger study.

The evidence of low GI foods on cognitive performance in children is rather inconclusive and acknowledged by authors. The cognitive outcomes were not a part of the current study. Therefore, recommend to modify the abstract in line 20-22 to limit the discussion to glycemic response and to refrain from commenting on the implication for cognitive outcomes in current study. 

Author Response

We wish to thank the reviewer for the helpful comments and suggestions, which we have addressed in the revision of the manuscript. More specifically, we have made the following changes:

  1. The reviewer suggested that “ it would be helpful to know the BMI percentile of the participants since that is recommended to be used in children”. We have added information about percentile range observed in sample; Page 3, Line 119.
  2. The reviewer commented that it would be helpful to provide a reference for the use of the 24 to 48 hour washout period between test sessions (Line 132). We have added a reference to demonstrate that this time frame has been used in previous studies (Page 3 Line 140), yet it is important to note here that there is no reference which defines the ‘gold standard’ of the duration of washout. This is also in alignment with methodological developments [Brouns et al. (2005). Glycaemic index methodology. Nutrition Research Reviews] and the international standard for determination of GI (ISO 26642) does not even refer to washout. The only requirement being that subjects are in a fasted state for both the reference and the test product(s). In our study the tested products were simple sugars, 100% available and glycaemia returns to baseline latest after 3h without having further transformations happening (i.e. no fiber fermentation or protein hydrolysis) and 2) the diet of the children was not fully controlled (i.e. other ingested food and nutrients than the tested sugars might have more impact in long term). However, children were tested after a a 12 hours overnight fasting that followed a standardized evening meal which provided 25-30% of daily energy intake and was low in slowly-digestible carbohydrates in order to control for second meal effects (Wolever et al. (1988) Second-meal effect: low-glycemic-index foods eaten at dinner improve subsequent breakfast glycemic response. Am J Clin Nutr.]
  3. As cognition was not assessed in our study, the reviewer suggested to modify the abstract accordingly. We have followed this recommendation and omitted any refence to implication for cognitive performance (Page 1, Line 21-22)

Round 2

Reviewer 1 Report

Dear authors, I've appreciated your kind answer.

Still I confirm what was my previous thaught. Said that, since I do agree that evidence is either way limited to date, and also the other reviewer was eiger to accept your manuscript, I've changed my indication and ok with publishing.

Best regards